# Towards Simplicity in Deep Reinforcement Learning: Streamlined Off-Policy Learning

## Abstract

The field of Deep Reinforcement Learning (DRL) has recently seen a surge in the popularity of maximum entropy reinforcement learning algorithms. Their popularity stems from the intuitive interpretation of the maximum entropy objective and their superior sample efficiency on standard benchmarks. In this paper, we seek to understand the primary contribution of the entropy term to the performance of maximum entropy algorithms. For the Mujoco benchmark, we demonstrate that the entropy term in Soft Actor Critic (SAC) principally addresses the bounded nature of the action spaces. With this insight, we show how streamlined algorithms without entropy maximization can match the performance of SAC. We also propose a simple non-uniform sampling method for selecting transitions from the replay buffer during training. We further show that the streamlined algorithm with the simple non-uniform sampling scheme outperforms SAC and achieves state-of-the-art performance on challenging continuous control tasks.

## 1 Introduction

Off-policy Deep Reinforcement Learning (RL) algorithms aim to improve sample efficiency by reusing past experience. Recently a number of new off-policy Deep Reinforcement Learning algorithms have been proposed for control tasks with continuous state and action spaces, including Deep Deterministic Policy Gradient (DDPG) and Twin Delayed DDPG (TD3) (Lillicrap et al., 2015; Fujimoto et al., 2018). TD3, which introduced clipped double-Q learning, delayed policy updates and target policy smoothing, has been shown to be significantly more sample efficient than popular on-policy methods for a wide range of Mujoco benchmarks.

The field of Deep Reinforcement Learning (DRL) has also recently seen a surge in the popularity of maximum entropy RL algorithms. Their popularity stems from the intuitive interpretation of the maximum entropy objective and their superior sample efficiency on standard benchmarks. In particular, Soft Actor Critic (SAC), which combines off-policy learning with maximum-entropy RL, not only has many attractive theoretical properties, but can also give superior performance on a wide-range of Mujoco environments, including on the high-dimensional environment Humanoid for which both DDPG and TD3 perform poorly (Haarnoja et al., 2018a;b; Langlois et al., 2019). SAC has a similar structure to TD3, but also employs maximum entropy reinforcement learning.

In this paper, we first seek to understand the primary contribution of the entropy term to the performance of maximum entropy algorithms. For the Mujoco benchmark, we demonstrate that when using the standard objective without entropy along with standard additive noise exploration, there is often insufficient exploration due to the bounded nature of the action spaces. Specifically, the outputs of the policy network are often way outside the bounds of the action space, so that they need to be squashed to fit within the action space. The squashing results in actions persistently taking on their maximal values, so that there is insufficient exploration. In contrast, the entropy term in the SAC objective forces the outputs to have sensible values, so that even with squashing, exploration is maintained. We conclude that the entropy term in the objective for Soft Actor Critic principally addresses the bounded nature of the action spaces in the Mujoco environments.

With this insight, we propose Streamlined Off Policy (SOP), a streamlined algorithm using the standard objective without the entropy term. SOP employs a simple normalization scheme to address the bounded nature of the action spaces, allowing satisfactory exploration throughout training. We also consider replacing the aforementioned normalization scheme with inverting gradients (IG)

Hausknecht & Stone (2015). Our results show that SOP and IG match the sample-efficiency and robustness performance of SAC, including on the more challenging Ant and Humanoid environments. This demonstrates a need to revisit the importance of entropy maximization in DRL.

Keeping with the theme of simplicity with the goal of meeting Occam's principle, we also propose a simple non-uniform sampling method for selecting transitions from the replay buffer during training. In vanilla SOP (as well as in DDPG, TD3, and SAC), samples from the replay buffer are chosen uniformly at random during training. Our method, called Emphasizing Recent Experience (ERE), samples more aggressively recent experience while not neglecting past experience. Unlike Priority Experience Replay (PER) (Schaul et al., 2015), a popular non-uniform sampling scheme for the Atari environments, ERE is only a few lines of code and does not rely on any sophisticated data structures. We show that SOP combined with ERE out-performs SAC and provides state of the art performance. For example, for Ant and Humanoid, it improves over SAC by $21\%$ and $24\%$, respectively, with one million samples. Furthermore, we also investigate combining SOP with PER, and show SOP+ERE also out-performs the more complicated SOP+PER scheme.

The contributions of this paper are thus threefold. First, we uncover the primary contribution of the entropy term of maximum entropy RL algorithms when the environments have bounded action spaces. Second, we propose a streamlined algorithm which do not employ entropy maximization but nevertheless matches the sampling efficiency and robustness performance of SAC for the Mujoco benchmarks. And third, we combine our streamlined algorithms with a simple non-uniform sampling scheme to achieve state-of-the art performance for the Mujoco benchmarks. We provide anonymized code for reproducibility [1].

## 2 PRELIMINARIES

We represent an environment as a Markov Decision Process (MDP) which is defined by the tuple $(\mathcal{S}, \mathcal{A}, r, p, \gamma)$, where $\mathcal{S}$ and $\mathcal{A}$ are continuous multi-dimensional state and action spaces, $r(s, a)$ is a bounded reward function, $p(s'|s, a)$ is a transition function, and $\gamma$ is the discount factor. Let $s(t)$ and $a(t)$ respectively denote the state of the environment and the action chosen at time $t$. Let $\pi = \pi(a|s)$, $s \in \mathcal{S}, a \in \mathcal{A}$ denote the policy. We further denote $K$ for the dimension of the action space, and write $a_k$ for the $k$th component of an action $a \in \mathcal{A}$, that is, $a = (a_1, \ldots, a_K)$.

The expected discounted return for policy $\pi$ beginning in state $s$ is given by:

$$V_\pi(s) = \mathbb{E}_\pi[\sum_{t=0}^{\infty} \gamma^t r(s(t), a(t))|s(0) = s] \tag{1}$$

Standard MDP and RL problem formulations seek to maximize $V_\pi(s)$ over policies $\pi$. For finite state and action spaces, under suitable conditions for continuous state and action spaces, there exists an optimal policy that is deterministic (Puterman, 2014; Bertsekas & Tsitsiklis, 1996). In RL with unknown environment, exploration is required to learn a suitable policy.

In DRL with continuous action spaces, typically the policy is modeled by a parameterized policy network which takes as input a state $s$ and outputs a value $\mu(s; \theta)$, where $\theta$ represents the current parameters of the policy network (Schulman et al., 2015; 2017; Vuong et al., 2018; Lillicrap et al., 2015; Fujimoto et al., 2018). During training, typically additive random noise is added for exploration, so that the actual action taken when in state $s$ takes the form $a = \mu(s; \theta) + \epsilon$ where $\epsilon$ is a $K$-dimensional Gaussian random vector with each component having zero mean and variance $\sigma$. During testing, $\epsilon$ is set to zero.

### 2.1 ENTROPY MAXIMIZATION RL

Maximum entropy reinforcement learning takes a different approach than (1) by optimizing policies to maximize both the expected return and the expected entropy of the policy (Ziebart et al., 2008; Ziebart, 2010; Todorov, 2008; Rawlik et al., 2013; Levine & Koltun, 2013; Levine et al., 2016; Nachum et al., 2017; Haarnoja et al., 2017; 2018a;b).

---

[1] https://anonymous.4open.science/r/e484a8c7-268a-4a66-a001-1e7676540237/

In particular, with maximization entropy RL, the objective is to maximize

$$V_\pi(s) = \sum_{t=0}^{\infty} \gamma^t \mathbb{E}_\pi [r(s(t), a(t)) + \lambda H(\pi(\cdot|s(t)))|s(0) = s] \tag{2}$$

where $H(\pi(\cdot|s))$ is the entropy of the policy when in state $s$, and the temperature parameter $\lambda$ determines the relative importance of the entropy term against the reward.

For entropy maximization DRL, when given state $s$ the policy network will typically output a $K$-dimensional vector $\sigma(s;\theta)$ in addition to the vector $\mu(s;\theta)$. The action selected when in state $s$ is then modeled as $\mu(s;\theta) + \epsilon$ where $\epsilon \sim N(0, \sigma(s;\theta))$.

Maximum entropy RL has been touted to have a number of conceptual and practical advantages for DRL (Haarnoja et al., 2018a;b). For example, it has been argued that the policy is incentivized to explore more widely, while giving up on clearly unpromising avenues. It has also been argued that the policy can capture multiple modes of near-optimal behavior, that is, in problem settings where multiple actions seem equally attractive, the policy will commit equal probability mass to those actions. In this paper, we show for the Mujoco benchmarks that the standard additive noise exploration suffices and can achieve the same performance as maximum entropy RL.

## 3 THE SQUASHING EXPLORATION PROBLEM

### 3.1 BOUNDED ACTION SPACES

Continuous environments typically have bounded action spaces, that is, along each action dimension $k$ there is a minimum possible action value $a_k^{\min}$ and a maximum possible action value $a_k^{\max}$. When selecting an action, the action needs to be selected within these bounds before the action can be taken. DRL algorithms often handle this by squashing the action so that it fits within the bounds. For example, if along any one dimension the value $\mu(s;\theta) + \epsilon$ exceeds $a_{\max}$, the action is set (clipped) to $a_{\max}$. Alternatively, a smooth form of squashing can be employed. For example, suppose $a_k^{\min} = -M$ and $a_k^{\max} = +M$ for some positive number $M$, then a smooth form of squashing could use $a = M \tanh(\mu(s;\theta) + \epsilon)$ in which $\tanh()$ is being applied to each component of the $K$-dimensional vector. DDPG (Hou et al., 2017) and TD3 (Fujimoto et al., 2018) use clipping, and SAC (Haarnoja et al., 2018a;b) uses smooth squashing with the $\tanh()$ function. For concreteness, henceforth we will assume that smooth squashing with the $\tanh()$ is employed.

We note that an environment may actually allow the agent to input actions that are outside the bounds. In this case, the environment will typically first clip the actions internally before passing them on to the "actual" environment (Fujita & Maeda, 2018).

We now make a simple but crucial observation: squashing actions to fit into a bounded action space can have a disastrous effect on additive-noise exploration strategies. To see this, let the output of the policy network be $\mu(s) = (\mu_1(s), \ldots, \mu_K(s))$. Consider an action taken along one dimension $k$, and suppose $\mu_k(s) >> 1$ and $|\epsilon_k|$ is relatively small compared to $\mu_k(s)$. Then the action $a_k = M \tanh(\mu_k(s) + \epsilon_k)$ will be very close (essentially equal) to $M$. If the condition $\mu_k(s) >> 1$ persists over many consecutive states, then $a_k$ will remain close to 1 for all these states, and consequently there will be essentially no exploration along the $k$th dimension. We will refer to this problem as the *squashing exploration problem*. A similar observation was made in Hausknecht & Stone (2015). We will argue that algorithms such as DDPG and TD3 based on the standard objective (1) with additive noise exploration can be greatly impaired by squashing exploration.

### 3.2 WHAT DOES ENTROPY MAXIMIZATION BRING TO SAC FOR THE MUJUCO ENVIRONMENTS?

SAC is a maximum-entropy based off-policy DRL algorithm which provides good performance across all of the Mujoco benchmark environments. To the best of our knowledge, it currently provides state of the art performance for the Mujoco benchmark. In this section, we argue that the principal contribution of the entropy term in the SAC objective is to resolve the squashing exploration problem, thereby maintaining sufficient exploration when facing bounded action spaces. To argue this, we consider two DRL algorithms: SAC with adaptive temperature (Haarnoja et al., 2018b), and

SAC with entropy removed altogether (temperature set to zero) but everything else the same. We refer to them as *SAC* and as *SAC without entropy*. For SAC without entropy, for exploration we use additive zero-mean Gaussian noise with $\sigma$ fixed at $0.3$. Both algorithms use $\mathtt{tanh}$ squashing. We compare these two algorithms on two Mujoco environments: Humanoid-v2 and Walker-v2.

Figure 1 shows the performance of the two algorithms with 10 seeds. For Humanoid, SAC performs much better than SAC without entropy. However, for Walker, SAC without entropy performs nearly as well as SAC, implying maximum entropy RL is not as critical for this environment.

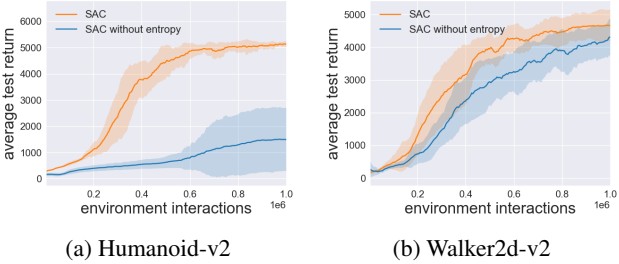

(a) Humanoid-v2     (b) Walker2d-v2

Figure 1: SAC performance with and without entropy maximization

To understand why entropy maximization is important for one environment but less so for another, we examine the actions selected when training these two algorithms. Humanoid and Walker have action dimensions $K = 17$ and $K = 6$, respectively. Here we show representative results for one dimension for both environments, and provide the full results in the Appendix. The top and bottom rows of Figure 2 shows results for Humanoid and Walker, respectively. The first column shows the $\mu_k$ values for an interval of 1,000 consecutive time steps, namely, for time steps 599,000 to 600,000. The second column shows the actual action values passed to the environment for these time steps. The third and fourth columns show a concatenation of 10 such intervals of 1000 time steps, with each interval coming from a larger interval of 100,000 time steps.

The top and bottom rows of Figure 2 are strikingly different. For Humanoid using SAC with entropy, the $|\mu_k|$ values are small, mostly in the range [-1.5,1.5], and fluctuate significantly. This allows the action values to also fluctuate significantly, providing exploration in the action space. On the other hand, for SAC without entropy the $|\mu_k|$ values are typically huge, most of which are well outside the interval [-10,10]. This causes the actions $a_k$ to be persistently clustered at either $M$ or $-M$, leading to essentially no exploration along that dimension. As shown in the Appendix, this property (lack of exploration for SAC without entropy maximization) holds for all 17 action dimensions. For Walker, we see that for both algorithms, the $\mu_k$ values are sensible, mostly in the range [-1,1] and therefore the actions chosen by both algorithms exhibit exploration.

In conclusion, the principle benefit of maximum entropy RL in SAC for the Mujuco environments is that it resolves the squashing exploration problem. For some environments (such as Walker), the outputs of the policy network take on sensible values, so that sufficient exploration is maintained and overall good performance is achieved without the need for entropy maximization. For other environments (such as Humanoid), entropy maximization is needed to reduce the magnitudes of the outputs so that exploration is maintained and overall good performance is achieved.

## 4 Streamlined Off-Policy (SOP) Algorithm

Given the observations in the previous section, a natural question is: is it possible to design a streamlined off policy algorithm that does not employ entropy maximization but offers performance comparable to SAC (which has entropy maximization)?

As we observed in the previous section, without entropy maximization, in some environments the policy network output values $|\mu_k|$, $k = 1, \ldots, K$ can become persistently huge, which leads to insufficient exploration due to the squashing. A simple solution is to modify the outputs of the policy network by normalizing the output values when they collectively (across the action dimensions) become too large. To this end, let $\mu = (\mu_1, \ldots, \mu_K)$ be the output of the original policy network, and let $G = \sum_k |\mu_k|/K$. The $G$ is simply the average of the magnitudes of the components of $\mu$. The

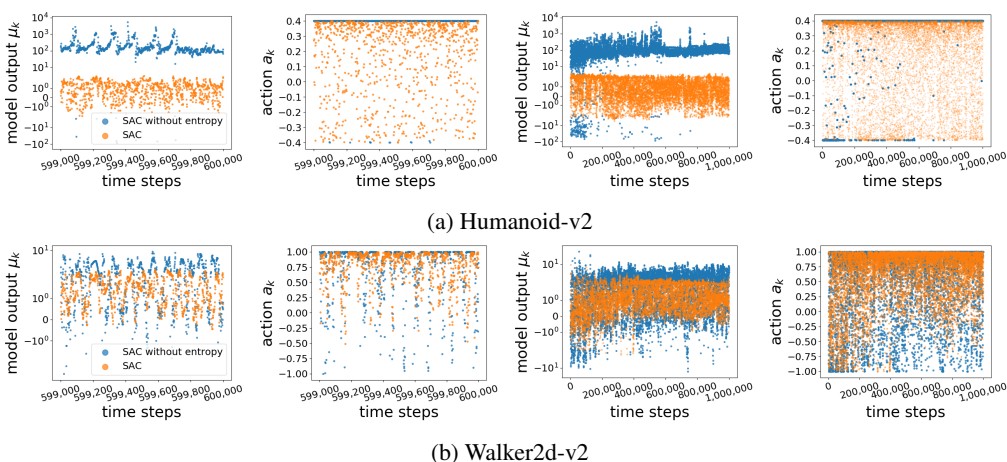

Figure 2: $\mu_k$ and $a_k$ values from SAC and SAC without entropy maximization

normalization procedure is as follows. If $G > 1$, then we reset $\mu_k \leftarrow \mu_k/G$ for all $k = 1, \ldots, K$; otherwise, we leave $\mu$ unchanged. With this simple normalization, we are assured that the average of the normalized magnitudes is never greater than one. Henceforth we assume the policy network has been modified with the simple normalization scheme just described.

Our Streamlined Off Policy (SOP) algorithm is described in Algorithm 1. The algorithm is essentially DDPG plus the normalization described above, plus clipped double Q-learning and target policy smoothing (Fujimoto et al., 2018). Another way of looking at it is as TD3 plus the normalization described above, minus the delayed policy updates and the target policy parameters. SOP also uses tanh squashing instead of clipping, since tanh gives somewhat better performance in our experiments. The SOP algorithm is "streamlined" as it has no entropy terms, temperature adaptation, target policy parameters or delayed policy updates. In our experiments, we also consider TD3 plus the simple normalization, and also another streamlined algorithm in which we replace the simple normalization scheme described above with the inverting gradients (IG) scheme as described in Hausknecht & Stone (2015). The basic idea is: when gradients suggest increasing the action magnitudes, gradients will be downscaled if actions are within the boundaries, and inverted entirely if actions are outside the boundaries. More implementation details can be found in the Appendix.

---

**Algorithm 1** Streamlined Off-Policy

1: Input: initial policy parameters $\theta$, Q-function parameters $\phi_1$, $\phi_2$, empty replay buffer $\mathcal{D}$
2: Set target parameters equal to main parameters $\phi_{\text{targ}_i} \leftarrow \phi_i$ for i = 1, 2
3: **repeat**
4:     Generate an episode using actions $a = M\text{tanh}(\mu_\theta(s) + \epsilon)$ where $\epsilon \sim \mathcal{N}(0, \sigma_1)$.
5:     **for** $j$ in range(however many updates) **do**
6:         Randomly sample a batch of transitions, $B = \{(s, a, r, s)\}$ from $\mathcal{D}$
7:         Compute targets for Q functions:
            $y_q(r, s') = r + \gamma \min_{i=1,2} Q_{\phi_{\text{targ}_i}}(s', M\text{tanh}(\mu_\theta(s') + \delta))$     $\delta \sim \mathcal{N}(0, \sigma_2)$
8:         Update Q-functions by one step of gradient descent using
            $\nabla_{\phi_i} \frac{1}{|B|} \sum_{(s,a,r,s') \in B} \left( Q_{\phi_i}(s, a) - y_q(r, s') \right)^2$ for $i = 1, 2$
9:         Update policy by one step of gradient ascent using
            $\nabla_\theta \frac{1}{|B|} \sum_{s \in B} Q_{\phi_1}(s, M\text{tanh}(\mu_\theta(s)))$
10:    Update target networks with
            $\phi_{\text{targ}_i} \leftarrow \rho \phi_{\text{targ}_i} + (1 - \rho) \phi_i$ for $i = 1, 2$

---

## 4.1 EXPERIMENTAL RESULTS FOR SOP

Figure 3 compares SAC (with temperature adaptation (Haarnoja et al., 2018a;b)) with SOP, TD3+ (that is, TD3 plus the simple normalization), and inverting gradients (IG) for five of the most chal-

lenging Mujuco environments. Using the same baseline code, we train with ten different random seeds for each of the two algorithms. Each algorithm performs five evaluation rollouts every 5000 environment steps. The solid curves correspond to the mean, and the shaded region to the standard deviation of the returns over the ten seeds.

Results show that SOP, SAC and IG have similar sample-efficiency performance and robustness across all environments. TD3+ has slightly weaker asymptotic performance for Walker and Humanoid. IG initially learns slowly for Humanoid with high variance across random seeds, but gives similar asymptotic performance. This confirms that with a simple output normalization scheme in the policy network, the performance of SAC can be achieved without maximum entropy RL.

In the Appendix we provide an ablation study for SOP, which shows a major performance drop when removing either double Q-learning or normalization, whereas removing target policy smoothing (Fujimoto et al., 2018) results in only a small performance drop in some environments.

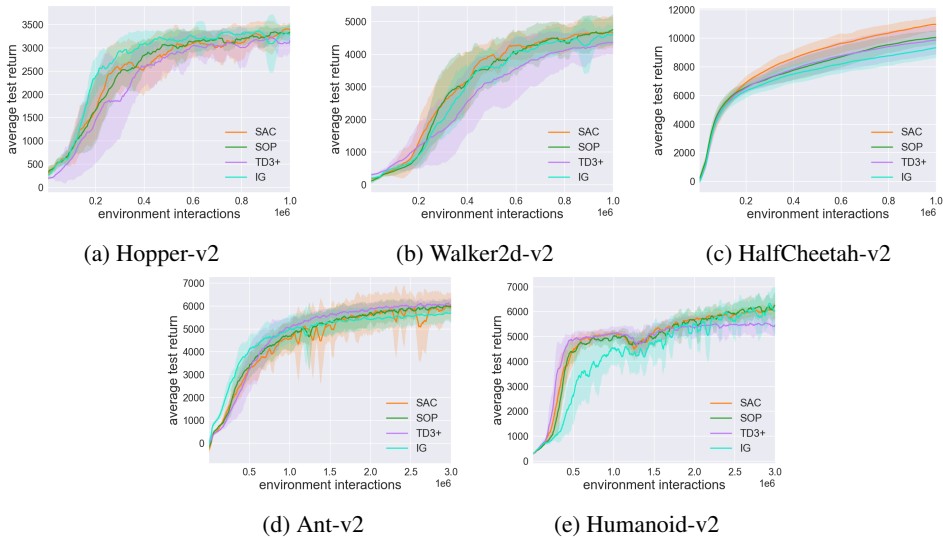

Figure 3: Streamlined Off-Policy (SOP) versus SAC, TD3+ and IG

# 5 NON-UNIFORM SAMPLING

We now show how a small change in the sampling scheme for SOP can achieve state of the art performance for the Mujoco benchmark. We call this sampling scheme Emphasizing Recent Experience (ERE). ERE has 3 core features: $(i)$ It is a general method applicable to any off-policy algorithm; $(ii)$ It requires no special data structure, is very simple to implement, and has near-zero computational overhead; $(iii)$ It only introduces one additional important hyper-parameter.

The basic idea is: during the parameter update phase, the first mini-batch is sampled from the entire buffer, then for each subsequent mini-batch we gradually reduce our range of sampling to sample more aggressively from more recent data. Specifically, assume that in the current update phase we are to make 1000 mini-batch updates. Let $N$ be the max size of the buffer. Then for the $k^{th}$ update, we sample uniformly from the most recent $c_k$ data points, where $c_k = N \cdot \eta^k$ and $\eta \in (0, 1]$ is a hyper-parameter that determines how much emphasis we put on recent data. $\eta = 1$ is uniform sampling. When $\eta < 1$, $c_k$ decreases as we perform each update. $\eta$ can made to adapt to the learning speed of the agent so that we do not have to tune it for each environment.

The effect of such a sampling formulation is twofold. The first is recent data have a higher chance of being sampled. The second is that we do this in an ordered way: we first sample from all the data in the buffer, and gradually shrink the range of sampling to only sample from the most recent data. This scheme reduces the chance of over-writing parameter changes made by new data with parameter changes made by old data (French, 1999; McClelland et al., 1995; McCloskey & Cohen, 1989; Ratcliff, 1990; Robins, 1995). This process allows us to quickly obtain new information

from recent data, and better approximate the value functions near recently-visited states, while still maintaining an acceptable approximation near states visited in the more distant past.

What is the effect of replacing uniform sampling with ERE? First note if we do uniform sampling on a fixed buffer, the expected number of times a data point is sampled is the same for all data points. Now consider a scenario where we have a buffer of size 1000 (FIFO queue), we collect one data at a time, and then perform one update with mini-batch size of one. If we start with an empty buffer and sample uniformly, as data fills the buffer, each data point gets less and less chance of being sampled. Specifically, over a period of 1000 updates, the expected number of times the $t$th data is sampled is: $1/t + 1/(t+1) + \cdots + 1/T$. Figure 4f shows the expected number of times a data is sampled as a function of its position in the buffer. We see that older data are expected to get sampled much more than newer data. This is undesirable because when the agent is improving and exploring new areas of the state space; new data points may contain more interesting information than the old ones, which have already been updated many times.

When we apply the ERE scheme, we effectively skew the curve towards assigning higher expected number of samples for the newer data, allowing the newer data to be frequently sampled soon after being collected, which can accelerate the learning process. In the Appendix, we provide further algorithmic detail and analysis on ERE, and compare ERE to two other sampling schemes: an exponential sampling scheme and Prioritized Experience Replay (Schaul et al., 2015).

## 5.1 EXPERIMENTAL RESULTS FOR SOP+ERE

Figure 4 compares the performance of SOP, SOP+ERE, SAC and SAC+ERE. With ERE, both SAC and SOP gain a significant performance improvement in all environments. SOP+ERE learns faster than SAC and vanilla SOP in all Mujoco environments. SOP+ERE also greatly improves overall performance for the two most challenging environments, Ant and Humanoid, and has the best performance for Humanoid. In table 1, we show the mean test episode return and std across 10 random seeds at 1M timesteps for all environments. The last column displays the percentage improvement of SOP+ERE over SAC, showing that SOP+ERE achieves state of the art performance. In Ant and Humanoid, SOP+ERE improves performance by 21% and 24% over SAC at 1 million timesteps, respectively. As for the std, SOP+ERE gives lower values, and for Humanoid a higher value.

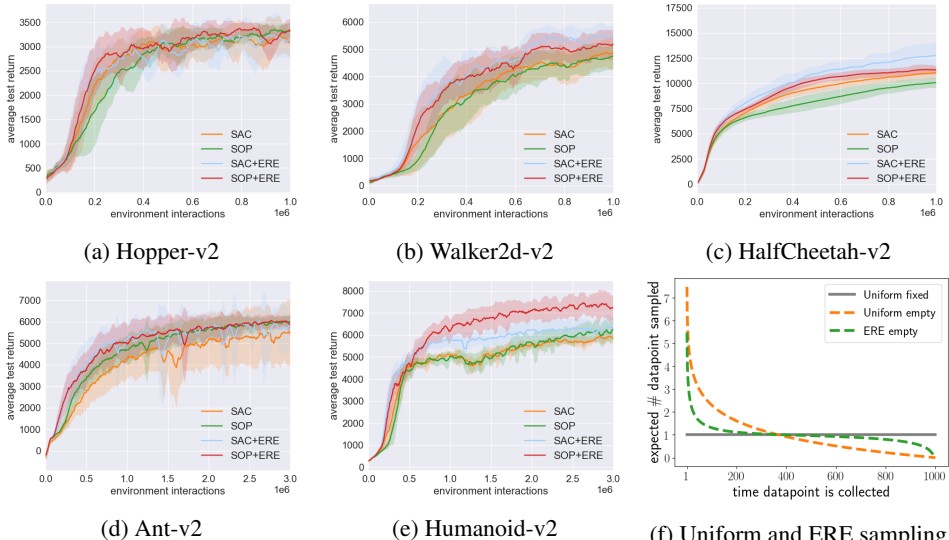

Figure 4: (a) to (e) show the performance of SOP and SAC with ERE sampling. (f) shows over a period of 1000 updates, the expected number of times the $t$th data point is sampled (with $\eta = 0.996$). ERE allows new data to be sampled many times soon after being collected.

Table 1: Performance comparison at one million samples. Last column shows percentage improvement of SOP+ERE over SAC.

| Environment | SAC Adaptive | SOP | SOP+ERE | Improvement |
|---|---|---|---|---|
| Hopper | $3161.2 \pm 381.0$ | $\mathbf{3317.3} \pm 133.9$ | $3201.5 \pm 248.7$ | 1.3% |
| Walker | $4801.5 \pm 514.5$ | $4666.5 \pm 474.5$ | $\mathbf{5145.9} \pm 512.3$ | 7.2% |
| HalfCheetah | $10963.7 \pm 512.4$ | $9968.0 \pm 497.4$ | $\mathbf{11335.1} \pm 478.3$ | 3.4% |
| Ant | $4153.7 \pm 925.0$ | $4674.0 \pm 588.8$ | $\mathbf{5023.3} \pm 891.6$ | 21.0% |
| Humanoid | $5076.2 \pm 148.1$ | $4900.9 \pm 316.6$ | $\mathbf{6297.7} \pm 500.0$ | 24.1% |

## 6 RELATED WORK

In recent years, there has been significant progress in improving the sample efficiency of DRL for continuous robotic locomotion tasks with off-policy algorithms (Lillicrap et al., 2015; Fujimoto et al., 2018; Haarnoja et al., 2018a;b). There is also a significant body of research on maximum entropy RL methods (Ziebart et al., 2008; Ziebart, 2010; Todorov, 2008; Rawlik et al., 2013; Levine & Koltun, 2013; Levine et al., 2016; Nachum et al., 2017; Haarnoja et al., 2017; 2018a;b). Ahmed et al. (2019) very recently shed light on how entropy leads to a smoother optimization landscape.

By taking clipping in the Mujoco environments explicitly into account, Fujita & Maeda (2018) modified the policy gradient algorithm to reduce variance and provide superior performance among on-policy algorithms. Eisenach et al. (2018) extend the work of Fujita & Maeda (2018) for when an action may be direction. Hausknecht & Stone (2015) introduce Inverting Gradients, for which we provide experimintal results in this paper for the Mujoco environments. Chou et al. (2017) also explores DRL in the context of bounded action spaces. Dalal et al. (2018) consider safe exploration in the context of constrained action spaces.

Uniform sampling is the most common way to sample from a replay buffer. One of the most well-known alternatives is prioritized experience replay (PER) (Schaul et al., 2015). PER uses the absolute TD-error of a data point as the measure for priority, and data points with higher priority will have a higher chance of being sampled. This method has been tested on DQN (Mnih et al., 2015) and double DQN (DDQN) (Van Hasselt et al., 2016) with significant improvement and applied successfully in other algorithms (Wang et al., 2015; Schulze & Schulze, 2018; Hessel et al., 2018; Hou et al., 2017) and can be implemented in a distributed manner (Horgan et al., 2018). There are other methods proposed to make better use of the replay buffer. The ACER algorithm has an on-policy part and an off-policy part, with a hyper-parameter controlling the ratio of off-policy to on-policy updates (Wang et al., 2016). The RACER algorithm (Novati & Koumoutsakos, 2018) selectively removes data points from the buffer, based on the degree of "off-policyness", bringing improvement to DDPG (Lillicrap et al., 2015), NAF (Gu et al., 2016) and PPO (Schulman et al., 2017). In De Bruin et al. (2015), replay buffers of different sizes were tested, showing large buffer with data diversity can lead to better performance. Finally, with Hindsight Experience Replay(Andrychowicz et al., 2017), priority can be given to trajectories with lower density estimation(Zhao & Tresp, 2019) to tackle multi-goal, sparse reward environments.

## 7 CONCLUSION

In this paper we first showed that the primary role of maximum entropy RL for the Mujoco benchmark is to maintain satisfactory exploration in the presence of bounded action spaces. We then developed a new streamlined algorithm which does not employ entropy maximization but nevertheless matches the sampling efficiency and robustness performance of SAC for the Mujoco benchmarks. Our experimental results demonstrate a need to revisit the benefits of entropy regularization in DRL. Finally, we combined our streamlined algorithm with a simple non-uniform sampling scheme to achieve state-of-the art performance for the Mujoco benchmark.

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

# A  ABLATION STUDY

In this ablation study we separately examine the importance of $(i)$ the normalization at the output of the policy network; $(ii)$ the double Q networks; $(iii)$ and randomization used in the line 8 of the SOP algorithm (that is, target policy smoothing (Fujimoto et al., 2018)).

Figure 5 shows the results for the five environments considered in this paper. In Figure 5, "no normalization" is SOP without the normalization of the outputs of the policy network; "single Q" is SOP with one Q-network instead of two; and "no smoothing" is SOP without the randomness in line 8 of the algorithm.

Figure 5 confirms that double Q-networks are critical for obtaining good performance (Van Hasselt et al., 2016; Fujimoto et al., 2018; Haarnoja et al., 2018a). Figure 5 also shows that output normalization is also critical. Without output normalization, performance fluctuates wildly, and average performance can decrease dramatically, particularly for Humanoid and HalfCheetah. Target policy smoothing improves performance by a relatively small amount.

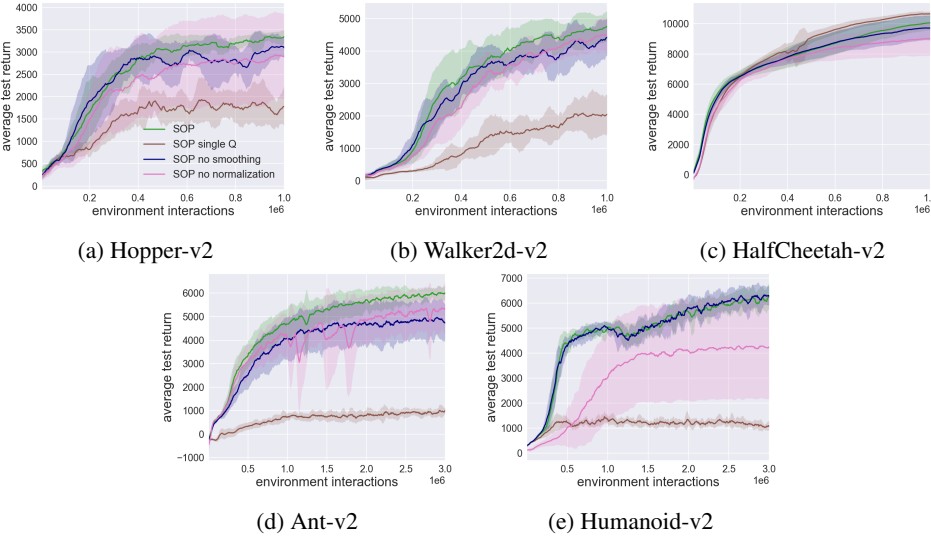

(a) Hopper-v2  (b) Walker2d-v2  (c) HalfCheetah-v2

(d) Ant-v2  (e) Humanoid-v2

Figure 5: Ablation Study

## B  HYPERPARAMETERS

Table 2 shows hyperparameters used for SOP, SOP+ERE and SOP+PER. For adaptive SAC, we use our own PyTorch implementation for the comparisons. Our implementation uses the same hyperparameters as used in the original paper (Haarnoja et al., 2018b). Our implementation of SOP variants and adaptive SAC share most of the code base. For TD3, our implementation uses the same hyperparamters as used in the authors' implementation, which is different from the ones in the original paper (Fujimoto et al., 2018). They claimed that the new set of hyperparamters can improve performance for TD3. We now discuss hyperparameter search for better clarity, fairness and reproducibility (Henderson et al., 2018; Duan et al., 2016; Islam et al., 2017).

For the $\eta$ value in the ERE scheme, in our early experiments we tried the values (0.993, 0.994, 0.995, 0.996, 0.997, 0.998) on the Ant and found 0.995 to work well. This initial range of values was decided by computing the ERE sampling range for the oldest data. We found that for smaller values, the range would simply be too small. For the PER scheme, we did some informal preliminary search, then searched on Ant for $\beta_1$ in (0, 0.4, 0.6, 0.8), $\beta_2$ in (0, 0.4, 0.5, 0.6, 1), and learning rate in (1e-4, 2e-4, 3e-4, 5e-4, 8e-4, 1e-3), we decided to search these values because the original paper used $\beta_1 = 0.6$, $\beta_2 = 0.4$ and with reduced learning rate. For the exponential sampling scheme, we searched the $\lambda$ value in (3e-7, 1e-6, 3e-6, 5e-6, 1e-5, 3e-5, 5e-5, 1e-4) in Ant, this search range was decided by plotting out the probabilities of sampling, and then pick a set of values that are not too extreme. For $\sigma$ in SOP, in some of our early experiments with SAC, we accidentally found that $\sigma = 0.3$ gives good performance for SAC without entropy and with Gaussian noise. We searched values (0.27, 0.28, 0.29, 0.3). For $\sigma$ values for TD3+, we searched values (0.1, 0.15, 0.2, 0.25, 0.3).

Table 2: SOP Hyperparameters

| Parameter | Value |
|---|---|
| *Shared* | |
|     optimizer | Adam (Kingma & Ba, 2014) |
|     learning rate | $3 \cdot 10^{-4}$ |
|     discount ($\gamma$) | 0.99 |
|     target smoothing coefficient ($\rho$) | 0.005 |
|     target update interval | 1 |
|     replay buffer size | $10^6$ |
|     number of hidden layers for all networks | 2 |
|     number of hidden units per layer | 256 |
|     mini-batch size | 256 |
|     nonlinearity | ReLU |
| *SAC adaptive* | |
|     entropy target | dim($\mathcal{A}$) (e.g., 6 for HalfCheetah-v2) |
| *SOP* | |
|     gaussian noise std $\sigma = \sigma_1 = \sigma_2$ | 0.29 |
| *TD3* | |
|     gaussian noise std for data collection $\sigma$ | 0.1 * action limit |
|     guassian noise std for target policy smoothing $\tilde{\sigma}$ | 0.2 |
| *TD3+* | |
|     gaussian noise std for data collection $\sigma$ | 0.15 |
|     guassian noise std for target policy smoothing $\tilde{\sigma}$ | 0.2 |
| *ERE* | |
|     ERE initial $\eta_0$ | 0.995 |
| *PER* | |
|     PER $\beta_1$ ($\alpha$ in PER paper) | 0.4 |
|     PER $\beta_2$ ($\beta$ in PER paper) | 0.4 |
| *EXP* | |
|     Exponential $\lambda$ | $5e - 06$ |

## C    ERE PSEUDOCODE

---

**Algorithm 2** SOP with Emphasizing Recent Experience

---

1: Input: initial policy parameters $\theta$, Q-function parameters $\phi_1$, $\phi_2$, empty replay buffer $\mathcal{D}$ of size $N$, initial $\eta_0$, recent and max performance improvement $I_{recent} = I_{max} = 0$.
2: Set target parameters equal to main parameters $\phi_{\text{targ,i}} \leftarrow \phi_i$  for i = 1, 2
3: **repeat**
4:     Generate an episode using actions $a = M\tanh(\mu_\theta(s) + \epsilon)$ where $\epsilon \sim \mathcal{N}(0, \sigma_1)$.
5:     update $I_{recent}, I_{max}$ with training episode returns, let $K =$ length of episode
6:     compute $\eta = \eta_0 \cdot \frac{I_{recent}}{I_{max}} + (1 - \frac{I_{recent}}{I_{max}})$
7:     **for** $j$ in range($K$) **do**
8:         Compute $c_k = N \cdot \eta^{k\frac{1000}{K}}$
9:         Sample a batch of transitions, $B = \{(s, a, r, s)\}$ from most recent $c_k$ data in $\mathcal{D}$
10:        Compute targets for Q functions:
             $$y_q(r, s') = r + \gamma \min_{i=1,2} Q_{\phi_{\text{targ},i}}(s', M\tanh(\mu_\theta(s') + \delta)) \quad \delta \sim \mathcal{N}(0, \sigma_2)$$
11:        Update Q-functions by one step of gradient descent using
             $$\nabla_{\phi_i} \frac{1}{|B|} \sum_{(s,a,r,s')\in B} (Q_{\phi,i}(s, a) - y_q(r, s'))^2 \text{ for } i = 1, 2$$
12:        Update policy by one step of gradient ascent using
             $$\nabla_\theta \frac{1}{|B|} \sum_{s\in B} Q_{\phi,1}(s, M\tanh(\mu_\theta(s)))$$
13:        Update target networks with
             $$\phi_{\text{targ, i}} \leftarrow \rho\phi_{\text{targ, i}} + (1 - \rho)\phi_i \text{ for } i = 1, 2$$

---

## D INVERTING GRADIENT METHOD

In this section we discuss the details of the Inverting Gradient method.

Hausknecht & Stone (2015) discussed three different methods for bounded parameter space learning: Zeroing Gradients, Squashing Gradients and Inverting Gradients, they analyzed and tested the three methods and found that Inverting Gradients method can achieve much stronger performance than the other two. In our implementation, we remove the tanh function from SOP and use Inverting Gradients instead to bound the actions. Let $p$ indicate the output of the last layer of the policy network. During exploration $p$ will be the mean of a normal distribution that we sample actions from, the IG approach can be summarized by the following equation (Hausknecht & Stone, 2015):

$$
\nabla_p = \nabla_p \cdot \begin{cases} (p_{\max} - p)/(p_{\max} - p_{\min}) & \text{if } \nabla_p \text{ suggests increasing } p \\ (p - p_{\min})/(p_{\max} - p_{\min}) & \text{otherwise} \end{cases} \tag{3}
$$

Where $\nabla_p$ is the gradient of the policy loss w.r.t to $p$. During a policy network update, we first backpropagate the gradients from the outputs of the Q network to the output of the policy network for each data point in the batch, we then compute the ratio $(p_{\max} - p)/(p_{\max} - p_{\min})$ or $(p_{\max} - p)/(p_{\max} - p_{\min})$ for each $p$ value (each action dimension), depending on the sign of the gradient. We then backpropagate from the output of the policy network to parameters of the policy network, and we modify the gradients in the policy network according to the ratios we computed. We made an efficient implementation and further discuss the computation efficiency of IG in the implementation details section.

## E SOP WITH OTHER SAMPLING SCHEMES

We also investigate the effect of other interesting sampling schemes.

### E.1 SAC WITH PRIORITIZED EXPERIENCE REPLAY

We also implement the proportional variant of Prioritized Experience Replay (Schaul et al., 2015) with SOP.

Since SOP has two Q-networks, we redefine the absolute TD error $|\delta|$ of a transition $(s, a, r, s')$ to be the average absolute TD error in the Q network update:

$$
|\delta| = \frac{1}{2} \sum_{l=1}^{2} |y_q(r, s') - Q_{\phi, l}(s, a)| \tag{4}
$$

Within the sum, the first term $y_q(r, s') = r + \gamma \min_{i=1,2} Q_{\phi_{\text{targ},i}}(s', \tanh(\mu_\theta(s') + \delta)), \delta \sim \mathcal{N}(0, \sigma_2)$ is simply the target for the Q network, and the term $Q_{\theta, l}(s, a)$ is the current estimate of the $l^{th}$ Q network. For the $i^{th}$ data point, the definition of the priority value $p_i$ is $p_i = |\delta_i| + \epsilon$. The probability of sampling a data point $P(i)$ is computed as:

$$
P(i) = \frac{p_i^{\beta_1}}{\sum_j p_j^{\beta_1}} \tag{5}
$$

where $\beta_1$ is a hyperparameter that controls how much the priority value affects the sampling probability, which is denoted by $\alpha$ in Schaul et al. (2015), but to avoid confusion with the $\alpha$ in SAC, we denote it as $\beta_1$. The importance sampling (IS) weight $w_i$ for a data point is computed as:

$$
w_i = \left( \frac{1}{N} \cdot \frac{1}{P(i)} \right)^{\beta_2} \tag{6}
$$

where $\beta_2$ is denoted as $\beta$ in Schaul et al. (2015).

Based on the SOP algorithm, we change the sampling method from uniform sampling to sampling using the probabilities $P(i)$, and for the Q updates we apply the IS weight $w_i$. This gives SOP with Prioritized Experience Replay (SOP+PER). We note that as compared with SOP+PER, ERE

does not require a special data structure and has negligible extra cost, while PER uses a sum-tree structure with some additional computational cost. We also tried several variants of SOP+PER, but preliminary results show that it is unclear whether there is improvement in performance, so we kept the algorithm simple.

### E.2 SOP WITH EXPONENTIAL SAMPLING

The ERE scheme is similar to an exponential sampling scheme where we assign the probability of sampling according to the probability density function of an exponential distribution. Essentially, in such a sampling scheme, the more recent data points get exponentially more probability of being sampled compared to older data.

For the $i^{th}$ most recent data point, the probability of sampling a data point $P(i)$ is computed as:

$$P(i) = \lambda e^{-\lambda x} \tag{7}$$

We apply this sampling scheme to SOP and refer to this variant as SOP+EXP.

### E.3 PER AND EXP EXPERIMENT RESULTS

Figure 6 shows a performance comparison of SOP, SOP+ERE, SOP+EXP and SOP+PER. Results show that the exponential sampling scheme gives a boost to the performance of SOP, and especially in the Humanoid environment, although not as good as ERE. Surprisingly, SOP+PER does not give a significant performance boost to SOP (if any boost at all). We also found that it is difficult to find hyperparameter settings for SOP+PER that work well for all environments. Some of the other hyperparameter settings actually reduce performance. It is unclear why PER does not work so well for SOP. A similar result has been found in another recent paper (Fu et al., 2019), showing that PER can significantly reduce performance on TD3. Further research is needed to understand how PER can be successfully adapted to environments with continuous action spaces and dense reward structure.

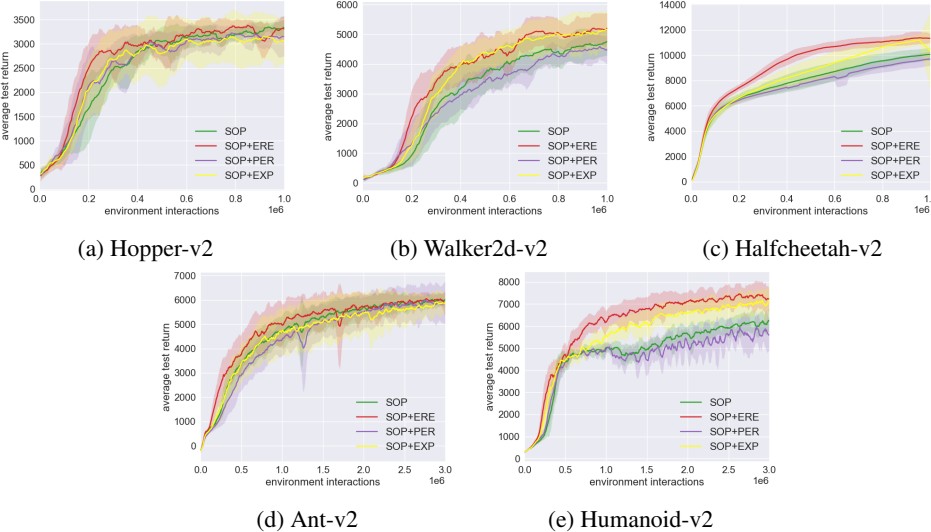

(a) Hopper-v2          (b) Walker2d-v2          (c) Halfcheetah-v2

(d) Ant-v2          (e) Humanoid-v2

Figure 6: Streamlined Off-Policy (SOP), with ERE and PER sampling schemes

## F ADDITIONAL ERE ANALYSIS

Figure 7 shows, for fixed $\eta$, how $\eta$ affects the data sampling process, under the ERE sampling scheme. Recent data points have a much higher probability of being sampled compared to older data, and a smaller $\eta$ value gives more emphasis to recent data.

Different $\eta$ values are desirable depending on how fast the agent is learning and how fast the past experiences become obsolete. So to make ERE work well in different environments with different reward scales and learning progress, we adapt $\eta$ to the the speed of learning. To this end, define performance to be the training episode return. Define $I_{recent}$ to be how much performance improved from $N/2$ timesteps ago, and $I_{max}$ to be the maximum improvement throughout training, where $N$ is the buffer size. Let the hyperparameter $\eta_0$ be the initial $\eta$ value. We then adapt $\eta$ according to the formula: $\eta = \eta_0 \cdot I_{recent}/I_{max} + 1 - (I_{recent}/I_{max})$.

Under such an adaptive scheme, when the agent learns quickly, the $\eta$ value is low in order to learn quickly from new data. When progress is slow, $\eta$ is higher to make use of the stabilizing effect of uniform sampling from the whole buffer.

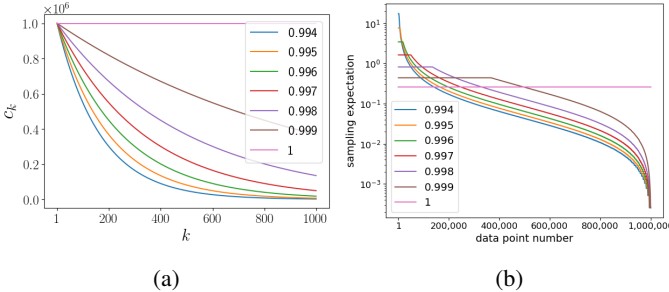

(a)                                        (b)

Figure 7: Effect of different $\eta$ values. The plots assume a replay buffer with 1 million samples, and 1,000 mini-batches of size 256 in an update phase. Figure 7a plots $c_k$ (ranging from 0 to 1 million) as a function of $k$ (ranging from 1 to 1,000). Figure 7b plots the expected number of times a data point in the buffer is sampled, with the data points ordered from most to least recent.

## G   ADDITIONAL IMPLEMENTATION DETAILS

### G.1   ERE IMPLEMENTATION

In this section we discuss some programming details. These details are not necessary for under-standing the algorithm, but they might help with reproducibility.

In the ERE scheme, the sampling range always starts with the entire buffer (1M data) and then gradually shrinks. This is true even when the buffer is not full. So even if there are not many data points in the buffer, we compute $c_k$ based as if there are 1M data points in the buffer. One can also modify the design slightly to obtain a variant that uses the current amount of data points to compute $c_k$. In addition to the reported scheme, we also tried shrinking the sampling range linearly, but it gives less performance gain.

In our implementation we set the number of updates after an episode to be the same as the number of timesteps in that episode. Since environments do not always end at 1000 timesteps, we can give a more general formula for $c_k$. Let $K$ be the number of mini-batch updates, let $N$ be the max size of the replay buffer, then:

$$c_k = N \cdot \eta^{k\frac{1000}{K}} \tag{8}$$

With this formulation, the range of sampling shrinks in more or less the same way with varying number of mini-batch updates. We always do uniform sampling in the first update, and we always have $\eta^{K\frac{1000}{K}} = \eta^{1000}$ in the last update.

When $\eta$ is small, $c_k$ can also become small for some of the mini-batches. To prevent getting a mini-batch with too many repeating data points, we set the minimum value for $c_k$ to 5000. We did not find this value to be too important and did not find the need to tune it. It also does not have any effect for any $\eta \geq 0.995$ since the sampling range cannot be lower than 6000.

In the adaptive scheme with buffer of size 1M, the recent performance improvement is computed as the difference of the current episode return compared to the episode return 500,000 timesteps earlier.

Before we reach 500,000 timesteps, we simply use $\eta_0$. The exact way of computing performance improvement does not have a significant effect on performance as long as it is reasonable.

### G.2 PROGRAMMING AND COMPUTATION COMPLEXITY

In this section we give analysis on the additional programming and computation complexity brought by ERE and PER.

In terms of programming complexity, ERE is a clear winner since it only requires a small adjustment to how we sample mini-batches. It does not modify how the buffer stores the data, and does not require a special data structure to make it work efficiently. Thus the implementation difficulty is minimal. PER (proportional variant) requires a sum-tree data structure to make it run efficiently. The implementation is not too complicated, but compared to ERE it is a lot more work.

The exponential sampling scheme is very easy to implement, although a naive implementation will incur a significant computation overhead when sampling from a large buffer. To improve its computation efficiency, we instead uses an approximate sampling method. We first sample data indexes from segments of size 100 from the replay buffer, and then for each segment sampled, we sample one data point uniformly from that segment.

In terms of computation complexity (not sample efficiency), and wall-clock time, ERE's extra computation is negligible. In practice we observe no difference in computation time between SOP and SOP+ERE. PER needs to update the priority of its data points constantly and compute sampling probabilities for all the data points. The complexity for sampling and updates is $O(log(N))$, and the rank-based variant is similar (Schaul et al., 2015). Although this is not too bad, it does impose a significant overhead on SOP: SOP+PER runs twice as long as SOP. Also note that this overhead grows linearly with the size of the mini-batch. The overhead for the Mujoco environments is higher compared to Atari, possibly because the Mujoco environments have a smaller state space dimension while a larger batch size is used, making PER take up a larger portion of computation cost. For the exponential sampling scheme, the extra computation is also close to negligible when using the approximate sampling method.

In terms of the proposed normalization scheme and the Inverting Gradients (IG) method, the normalization is very simple and can be easily implemented and has negligible computation overhead. IG has a simple idea, but its implementation is slightly more complicated than the normalization scheme. When implemented naively, IG can have a large computation overhead, but it can be largely avoided by making sure the gradient computation is still done in a batch-manner. We have made a very efficient implementation and our code is publicly available so that interested reader can easily reproduce it.

## H    ADDITIONAL EXPERIMENTAL RESULTS

### H.1    INVERTING GRADIENTS WITH ERE

In Figure 8 we show additional results on applying ERE to SOP+IG. The result shows that after applying the ERE scheme, SOP and IG both get a performance boost. The performance of the SOP+ERE and IG+ERE are similar.

### H.2    TD3 VERSUS TD3+

In figure 9, we show additional results comparing TD3 with TD3 plus our normalization scheme, which we refer as TD3+. The results show that after applying our normalization scheme, TD3+ has a significant performance boost in Humanoid, while in other environments, both algorithms achieve similar performance.

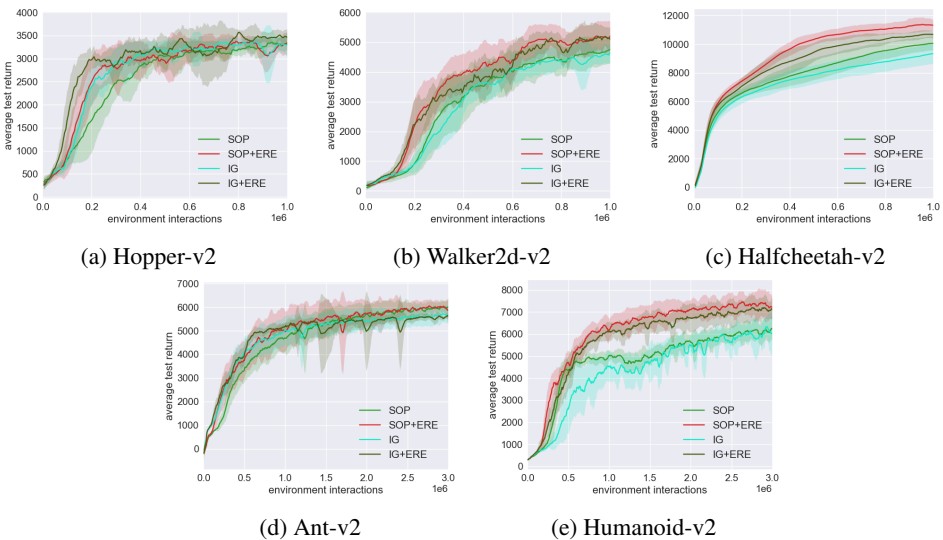

(a) Hopper-v2      (b) Walker2d-v2      (c) Halfcheetah-v2

(d) Ant-v2      (e) Humanoid-v2

Figure 8: SOP and inverting gradients with ERE sampling scheme

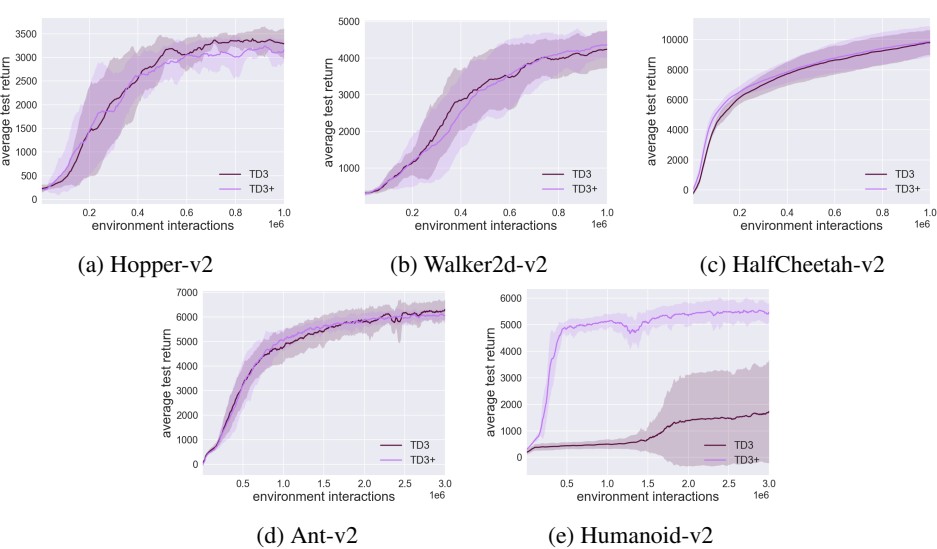

(a) Hopper-v2      (b) Walker2d-v2      (c) HalfCheetah-v2

(d) Ant-v2      (e) Humanoid-v2

Figure 9: TD3 versus TD3+ (TD3 plus the normalization scheme)

## I    ADDITIONAL ANALYSIS AND RESULTS COMPARING SAC WITH AND WITHOUT ENTROPY

To understand why entropy maximization is important for one environment but less so for another, we examine the actions selected when training SAC with and without entropy. Humanoid and Walker2d have action dimensions $K = 17$ and $K = 6$, respectively. In addition to the representative results shown for one dimension for both environments in Section 3.2, the results for all the dimensions are provided here in Figures 10 and 11.

From Figure 10, we see that for Humanoid using SAC (which uses entropy maximization), the $|\mu_k|$ values are small and fluctuate significantly for all 17 dimensions. On the other hand, for SAC without entropy the $|\mu_k|$ values are typically huge, again for all 17 dimensions. This causes the actions $a_k$ to be persistently clustered at either $M$ or -$M$. As for Walker, the $|\mu_k|$ values are sensible for both algorithms for all 6 dimensions, as shown in figure 11.

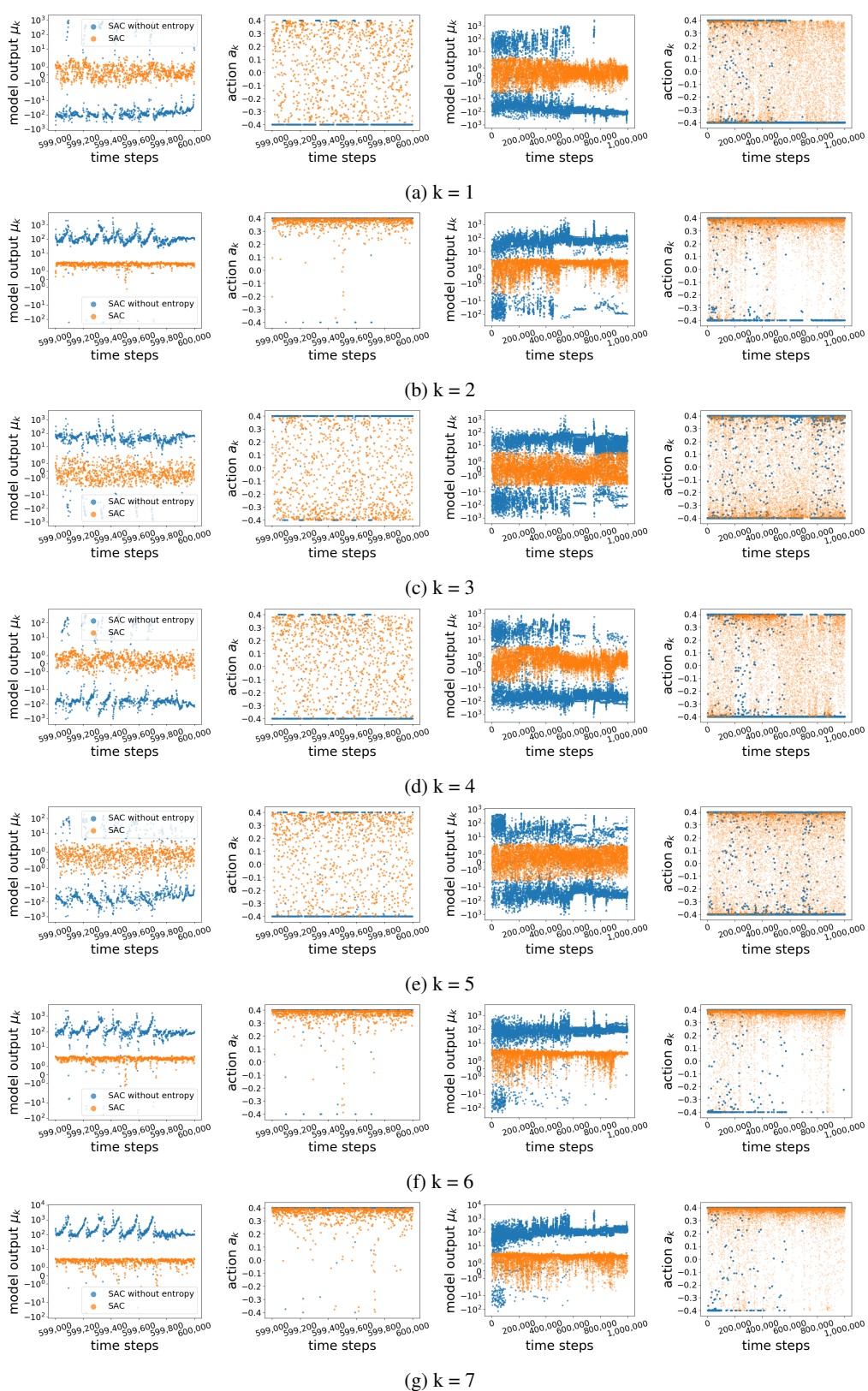

Figure 10: Humanoid-v2: $\mu_k$ and $a_k$ values from SAC and SAC without entropy maximization

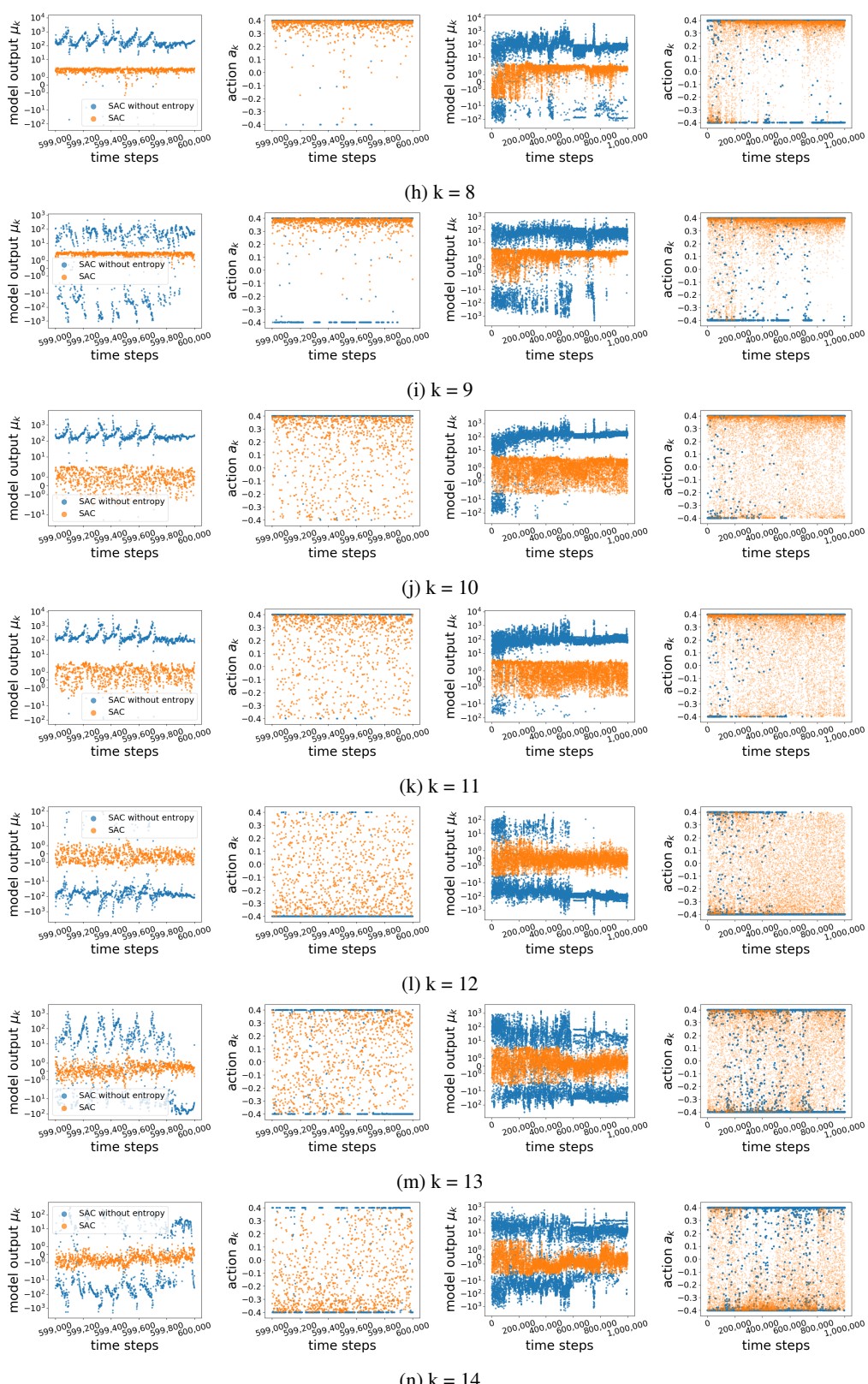

Figure 10: Humanoid-v2: $\mu_k$ and $a_k$ values from SAC and SAC without entropy maximization

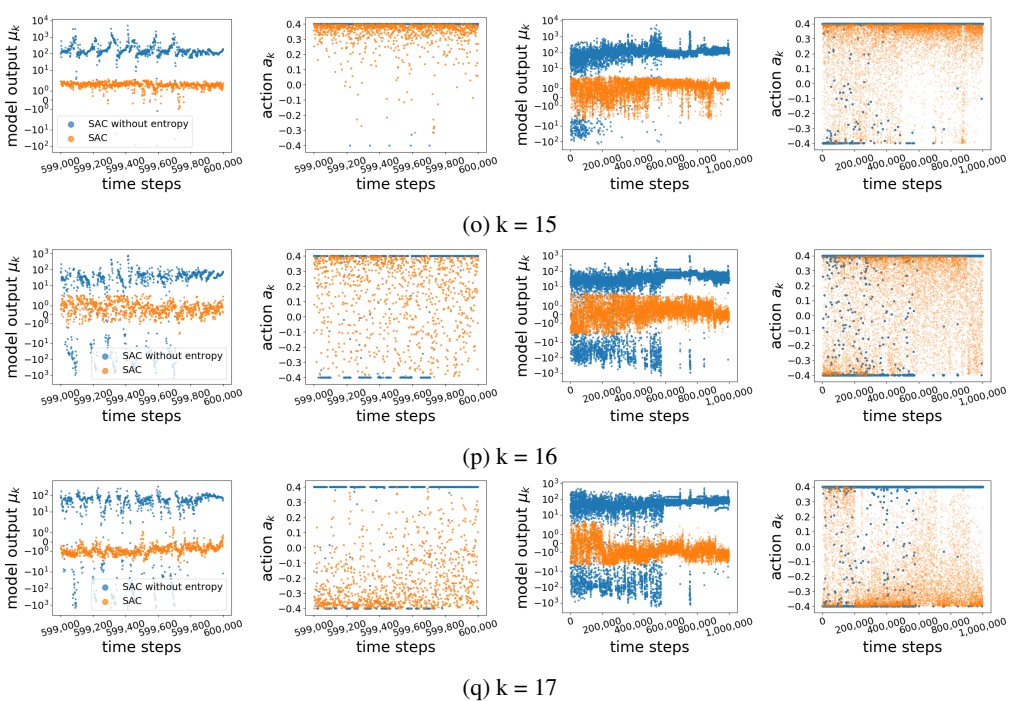

Figure 10: Humanoid-v2: $\mu_k$ and $a_k$ values from SAC and SAC without entropy maximization

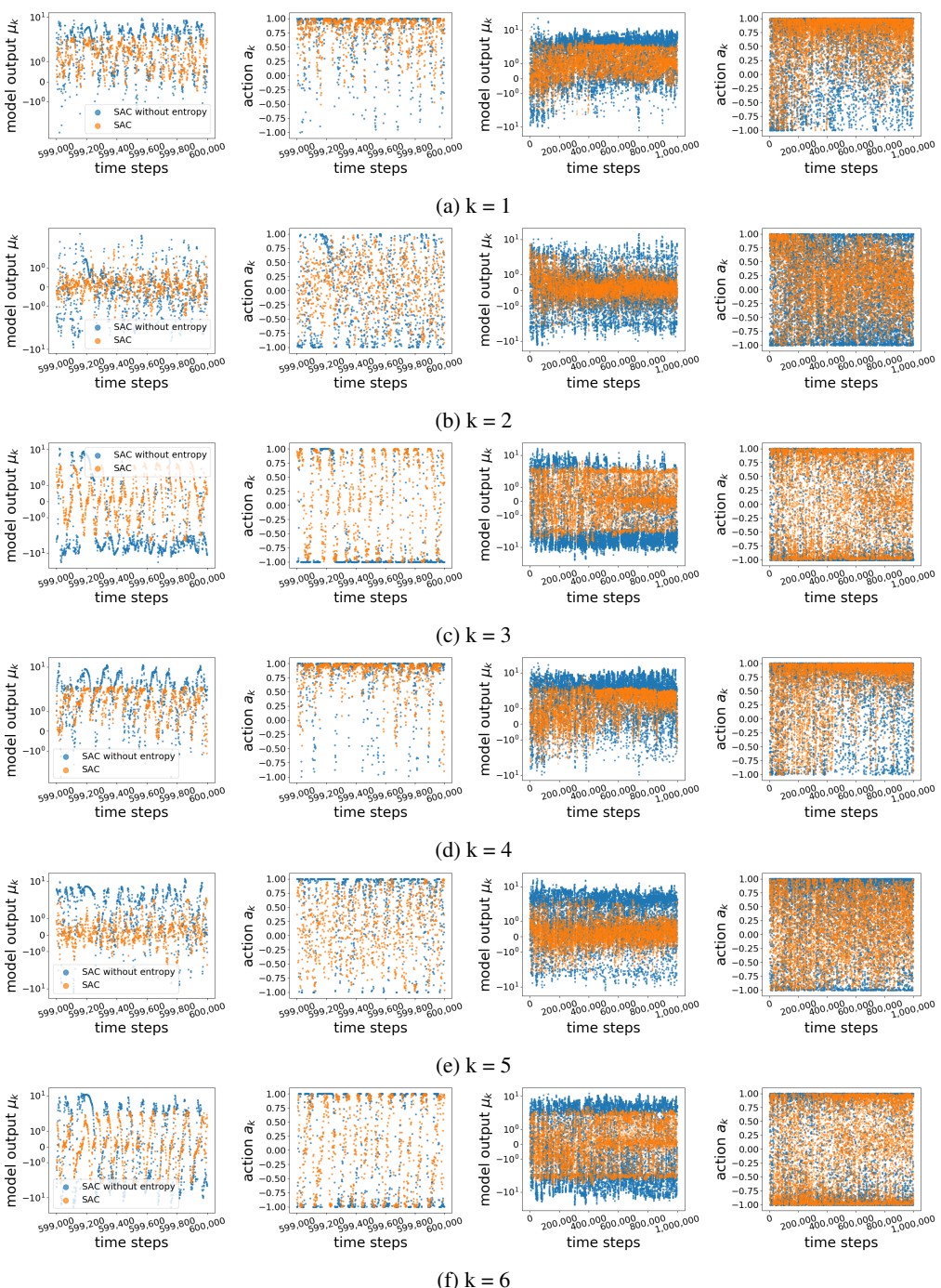

Figure 11: Walker2d-v2: $\mu_k$ and $a_k$ values from SAC and SAC without entropy maximization

