# OpenReview forum: "Towards Simplicity in Deep Reinforcement Learning: Streamlined Off-Policy Learning"
_ICLR.cc/2020/Conference — Reject_

### Official Review · AnonReviewer3 · 2019-10-07
**Official Blind Review #3**

**Rating:** 3

**Review:**

The authors investigate the role of entropy maximization in SAC and show that entropy regularization does not do what is usually thought: in the examples they investigate, where the output of the policy network needs to be squashed to fit in the action space domain, squashing would result in having only action at the boundaries, but entropy regularization maintains some intermediate values, hence exploration. From this insight, the authors replace entropy regularization by a simpler normalization process and show equivalent performance with their simpler Streamlined Off-Policy (SOP) algorithm. Then they introduce a second "Emphasizing Recent Experience" mechanism and show that SOP+ERE performs better than SAC.

A good point for the paper is that the entropy regularization  study is very nice, more papers in the field should show similar detailed analyses of internal processes. But the paper suffers from a few serious weaknesses:

- The TD3 mechanism goes beyond the Double Q-learning (or DDQN) mechanism of Van Hasselt et al: it takes the min over two critics. This should be explained properly.
- the title, abstract and introduction insist more on SOP, but performance improvement seem to result more from ERE. If this is possible, studying the performance of SAC + ERE would disambiguate the relative contribution of both mechanisms.

About gradient squashing issues, the authors main mention de gradient inverter idea from this paper:

@article{hausknecht2015deep,
  title={Deep reinforcement learning in parameterized action space},
  author={Hausknecht, Matthew and Stone, Peter},
  journal={arXiv preprint arXiv:1511.04143},
  year={2015}
}

The authors should also probably also cite (and read the latest arxiv version of):
@inproceedings{ahmed2019understanding,
  title={Understanding the impact of entropy on policy optimization},
  author={Ahmed, Zafarali and Le Roux, Nicolas and Norouzi, Mohammad and Schuurmans, Dale},
  booktitle={International Conference on Machine Learning},
  pages={151--160},
  year={2019}
}


More local points:
- "without performing a careful hyper-parameter search": so how did you choose these hyper-parameters? I see what you mean, but this is a very vague and slippery statement.
- I do not find the 23*4 images in Appendix B much useful
- Fig 3 seems to be repeated in Fig 4. Can't you just remove Fig 3?

**Experience Assessment:**

I have read many papers in this area.

**Review Assessment: Checking Correctness Of Derivations And Theory:**

I did not assess the derivations or theory.

**Review Assessment: Checking Correctness Of Experiments:**

I assessed the sensibility of the experiments.

**Review Assessment: Thoroughness In Paper Reading:**

I read the paper at least twice and used my best judgement in assessing the paper.

---

> ### Author Response · Authors · 2019-11-08
> **Reply to your comments**
>
> Thank you for your careful review of the paper. Your comments and suggestions are very useful. Below we respond to your various points.
>
> "The TD3 mechanism goes beyond the Double Q-learning (or DDQN) mechanism of Van Hasselt et al: it takes the min over two critics. This should be explained properly." Yes, thank you for bringing this to our attention. We will make this clear in the revision.
>
> "The title, abstract and introduction insist more on SOP, but performance improvement seem to result more from ERE. If this is possible, studying the performance of SAC + ERE would disambiguate the relative contribution of both mechanisms."  This is a good point. We are now running experiments for SAC+ERE, and we will include the results in the revision.
>
> "About gradient squashing issues, the authors main mention de gradient inverter idea from this paper. " We will discuss this paper in more detail in the revision. We will also cite the "Understanding the impact of entropy..." paper in the revision. Thank you for bringing it to our attention.
>
> We will respond to your "local points" in the revision, including a more thorough discussion of how the hyper-parameters were determined.

---

> ### Author Response · Authors · 2019-11-14
> **Reply to your review**
>
> Thank you for your comments. We have uploaded a new version of our paper and will continue working on it. Below is a summary of the changes we made so far:
>
> We now have a more detailed explanation of how SAC and TD3 are different. We explained that TD3 takes the min of 2 target critic values when computing update targets, and also added that TD3 uses delayed policy update and target policy smoothing.
>
> In the revision, we discuss the paper “Deep reinforcement learning in parameterized action space” in more detail and we also implemented the Inverting Gradients (IG) technique mentioned in the paper and compare its performance with SAC and SOP in figure 3. The IG experiments for Humanoid and Ant have not finished running yet. We expect them to finish tomorrow and we will update the figure once we have them. that IG has similar overall performance compared to SOP and SAC. It learns faster in the beginning in Hopper and Ant, but is slightly weaker in HalfCheetah, and did not do as well in Humanoid.
>
> We also include our results for SAC+ERE in figure 4, in comparison to the performance of SAC, SOP, and SOP+ERE. From the results, we can see that with ERE, both SAC and SOP gain a significant performance improvement in all environments. Since figure 3 and 4 now contain entirely different information, there is no redundancy.
>
> Moreover, we also include results for IG+ERE and compare it with IG, SOP and SOP+ERE in Appendix G (figure 8). Some IG+ERE experiments are still running. We will update the figure once we have them.
>
> We have read and cited the paper “Understanding the impact of entropy on policy optimization”. Thank you again for recommending this very insightful paper.
>
> Instead of a rather vague claim on hyper-parameters, we have now added a thorough discussion of exactly how we performed hyper-parameter search and why we chose those values in Appendix B Hyper-parameters section.
>
> The 23*4 images in the Appendix do take a lot of space, however, we are worried that other people may want to see the mu and action values for all action dimensions instead of just one action dimension we picked out in the main body. Thus, we decide to keep the 23*4 images but removed them to the end of the Appendix.

---

> ### Author Response · Authors · 2019-11-15
> **Final revision finished**
>
> We have just finished uploading a final revision of the paper. We have addressed all the "serious weaknesses" you mentioned in your comments, as well as the minor points. We have also added some other new experiments on Inverting Gradients and an exponential sampling scheme due to the request of other reviewers. A summary of the changes has been posted to all reviewers.
> We would like to thank you again for your helpful feedback, we feel that they helped us greatly in making this paper more refined. Thanks!

---

### Official Review · AnonReviewer2 · 2019-10-23
**Official Blind Review #2**

**Rating:** 6

**Review:**


# Summary
The paper identifies a problem with TD3 related to action clipping. The authors notice that SAC alleviates this problem by means of entropy regularization. Given the insight that action clipping is crucial, the authors propose an alternative approach to avoid action clipping in TD3, which is empirically shown to yield the same results as SAC. Surprisingly, with this improvement, even several parts from TD3 can be removed, such as delayed policy updates and the target policy parameters. In addition, a straightforward-to-implement experience replay scheme is proposed, that emphasizes recent experiences, which propels the proposed algorithm to achieve state-of-the-art results on MuJoCo.

# Decision
This is a great paper: accept. The proposed Streamlined Off Policy (SOP) algorithm is thoroughly evaluated, ablation studies performed, code made available. Nevertheless, there are a few suggestions below that may further improve the paper.

# Suggestions
1) It is said that entropy regularization leads to action not being saturated in SAC. I feel that this causal relation is very indirect. Maybe SAC with entropy just discovers better policies that do not go crazy between extremes? For example, if you would leave the entropy term but remove tanh saturation from SAC, don't you think you would also get a bang-bang policy? Adding such an ablation study could further strengthen the argument that entropy leads to no constraint violation, if it turns out true.

2) The Emphasizing Recent Experience (ERE) replay scheme seems reminiscent of sampling according to a distribution exponentially decaying into the past. It is said that physically shrinking the allowed sampling range by dropping old experiences is better because then very old experiences cannot be used at all. It would be interesting to see a comparison to sampling according to exponential distribution from the replay queue.

# AFTER REBUTTAL
Taking into account the concerns of other reviewers and the newly added evaluations, I lower my score to weak accept. Since now it seems that ERE is quite crucial, the argument of SPO outperforming SAC becomes weaker. Therefore, the authors should tone down the claims of outperforming SAC. Nevertheless, I still find the contribution of the paper valuable and think that it should be accepted, albeit with the aforementioned modifications in the camera-ready version.

**Experience Assessment:**

I have published one or two papers in this area.

**Review Assessment: Checking Correctness Of Derivations And Theory:**

I assessed the sensibility of the derivations and theory.

**Review Assessment: Checking Correctness Of Experiments:**

I assessed the sensibility of the experiments.

**Review Assessment: Thoroughness In Paper Reading:**

I read the paper at least twice and used my best judgement in assessing the paper.

---

> ### Author Response · Authors · 2019-11-08
> **Reply to your comments.**
>
> Thank you  for your careful review of the paper. We are happy that you think the paper is "great". We also feel that it makes an important contribution both in terms in bringing insight to off-policy DRL and achieving the state-of-art performance.
>
> We like your suggestion of comparing ERE with exponential sampling. We are currently running experiments, and we will include the results of these new experiments in the paper.
>
> Concerning your first suggestion, we would like to ask for a clarification. If we remove the entropy term from SAC, we found that the pre-tanh values become huge in magnitude, which leads to tanh saturation and poor exploration.  For the additional ablation study, do you mean to keep the entropy term, but then to remove tanh entirely (so that in some cases, the chosen action will be outside the bound), or do you mean keep the term and adding L1 penalty to the pre-tanh value?

---

> > ### Comment · AnonReviewer2 · 2019-11-12
> > **Remove tanh but keep the entropy penalty**
> >
> > Thanks for agreeing to do these additional experiments.
> >
> > I meant keeping the entropy term but remove tanh.

---

> > > ### Author Response · Authors · 2019-11-14
> > > **Re:Reply to your review**
> > >
> > > Thank you for the clarification! We have just uploaded a revision, here are some of the changes we made:
> > >
> > > We made an efficient implementation of the exponential sampling scheme (EXP) for SOP and performed experiments, the results are now reported in section D of the appendix, where we compare the performance of SOP with three sampling schemes, ERE, PER and EXP. The results show that EXP improves the performance of SOP consistently, and does well especially in the HalfCheetah and Humanoid environments, although the performance is not as strong as ERE.
> > >
> > > We have also included details on implementation and hyper-parameter search in the appendix. It turns a naive EXP sampling implementation will incur a significant computation overhead, but we avoided it by first sample segments of size 100 from the buffer, then sample a data uniformly from each segment. This modification does not really change the sampling scheme, is relatively easy to implement and has a negligible computation overhead.
> > >
> > > We also performed experiments on removing the tanh from SAC. However, our results show that if we simply remove tanh, it causes SAC performance to drop significantly. We suspect that some other parts of SAC will have to be modified in order for it to work correctly without tanh, but currently, it is unclear what is the missing part. We will try to continue work on this and run more experiments. And we can later add our findings to the camera-ready version.

---

### Official Review · AnonReviewer1 · 2019-10-23
**Official Blind Review #1**

**Rating:** 3

**Review:**

The main contribution of this paper is a normalization scheme to avoid saturating the squashing function typically used to constrain actions within a bounded range in continuous control problems. It is argued that algorithms like DDPG and TD3 suffer from such saturation, which prevents proper exploration during training, while maximum entropy algorithms like Soft Actor-Critic (SAC) avoid it thanks to their entropy bonus. The main reason behind the success of SAC would thus be its ability to keep exploring throughout training, by avoiding saturation. A second contribution is a new experience replay sampling scheme, named Emphasizing Recent Experience (ERE), based on the idea that most recently added transitions should be given higher weights when sampling mini-batches from the replay bufffer. Combining both ideas leads to the SOP (Streamlined Off-Policy)+ERE algorithm, which is shown to consistently outperform SAC on Mujoco tasks.

Although this paper presents interesting insights and very good empirical results, I am currently leaning towards rejection mostly due to missing some important empirical comparisons, which hopefully can be added in a revised version.

The first key missing comparison (IMO) is to the Inverting Gradients approach from Hausknecht & Stone (2016), which the authors know about since it is cited in the related work section. Note that in that paper, the problem of saturating squashing functions preventing proper exploration was already mentioned, although not investigated in as much depth as in this submission («(…) squashing functions quickly became saturated. The resulting agents take the same discrete action with the same maximum/minimum parameters each timestep »). Their proposed Inverting Gradients technique was found to work significantly better than squashing functions, which is why I believe it should be an obvious baseline to compare to.

The other important experiments which I think need to be added are simply to implement the proposed normalization scheme within DDPG & TD3 to demonstrate its usefulness as a standalone improvement over existing algorithms. This would strengthen the claim that « algorithms such as DDPG and TD3 based on the standard objective with additive noise exploration can be greatly impaired by squashing exploration ». Without this comparison on the same benchmark, it is difficult to fully grasp the impact of this normalization.

Finally, regarding the ERE sampling scheme, I would appreciate to see SAC+ERE as well, to (hopefully) show that it can benefit SAC too (since this second contribution is orthogonal to the SOP algorithm).

Minor points:
•	I would tone down a bit the claims for « the need to revisit the benefits of entropy maximization in DRL », since better exploration has always been put forward as a major benefit (« the maximum entropy formulation provides a substantial improvement in exploration and robustness », as written in « Soft Actor-Critic Algorithms and Applications »). To me, what this submission shows is essentially that naive implementation of additive noise exploration in e.g. DDPG is very bad for exploration, more than uncovering some novel properties of SAC.
•	Below eq. 1: « the optimal policy is deterministic » => should be replaced with « there exists an optimal policy that is deterministic »
•	« principle contribution » => principal
•	The normalization scheme does not appear in Alg. 1
•	In Alg. 1 there are a Q_phi,i and a Q_phi,1 that should probably be Q_phi_i and Q_phi_1
•	The results from section E in the Appendix should be mentioned in the main text
•	In Fig. 4f the y axis’ label is a bit clipped

Update based on new revision: thank you for adding more results. From what I can see, it is difficult to conclude on the benefits of SOP over IG, which I find really problematic. It seems to me that the most impactful result is related to the improvements brought by the ERE sampling scheme, which could probably be worth a paper on its own (by showing its benefits over a wider range of algorithms, e.g. TD3 & DQN+variants), but this would be a different paper. As a result I am sticking to "Weak Reject".

**Experience Assessment:**

I have read many papers in this area.

**Review Assessment: Checking Correctness Of Derivations And Theory:**

N/A

**Review Assessment: Checking Correctness Of Experiments:**

I carefully checked the experiments.

**Review Assessment: Thoroughness In Paper Reading:**

I read the paper thoroughly.

---

> ### Author Response · Authors · 2019-11-08
> **Reply to your review**
>
> Thank you for your careful reading of the paper. You have made many good suggestions, most of which we will address in the revision.
>
> As you suggest, we will implement the inverted gradient technique and compare the results with SOP and SAC. We will also be sure to acknowledge more fully the insights that paper made regarding saturating squashing functions.
>
> As you suggest, we are also running experiments for TD3+normalization. We agree in hindsight that this is a very natural combination to consider. Our results so far seem to indicate that for humanoid, normalization does improve the performance of TD3, but does not bring the performance to the level of SOP. We will provide the experimental results for all five environments in the revision.
>
> Reviewer 3 also pointed out that it would be good to see experimental results for SAC+ERE. We are currently running these experiments and will provide the results in the revision.
>
> We will also take care of your minor points in the revision. However, we do not fully agree with the first minor point. We feel that an important contribution of the paper is to show that entropy maximization is not a major benefit for the Mujoco environments; in fact by introducing a simple normalization of the outputs, we can achieve equivalent performance without entropy maximization.

---

> > ### Comment · AnonReviewer1 · 2019-11-12
> > **Re: Reply to your review**
> >
> > Thank you for running these additional experiments. If you can share their results before the end of the discussion period, that would be great.
> >
> > Regarding my first minor point, I guess this is mostly a matter of interpretation of your words. I understood them myself as "the reasons why SAC works so well are not those most people think", and this is something I tend to disagree with since what you show is essentially that SAC is better at exploring, and better exploration is a key motivation for the entropy maximization in SAC.

---

> > > ### Author Response · Authors · 2019-11-14
> > > **Re: Reply to your review**
> > >
> > > We have just uploaded a new version of our paper and will continue working on it.
> > > For the experiments you suggested,  below is a summary of what we did so far:
> > >
> > > In the revision, we implemented the Inverting Gradients(IG) technique and TD3+normalization, and compared their performance with SAC and SOP in figure 3. The IG experiments for Humanoid and Ant have not finished running yet. We expect them to finish tomorrow and we will update the figures once we have them. Our results show that IG has similar overall performance compared to SOP and SAC. It learns faster in the beginning in Hopper and Ant, but is slightly weaker in HalfCheetah, and did not do as well in Humanoid. TD3+normalization also has good performance, although not quite as good as the other schemes.
> > >
> > > We also compare the performance of TD3 with TD3+normalization. The results are shown in Appendix G (figure 9). Our results indicate that for humanoid, normalization boosts the performance of TD3 significantly, but does not bring the performance to the level of SOP.
> > >
> > > We also include our results for SAC+ERE in figure 4, in comparison to the performance of SAC, SOP, and SOP+ERE. From the results, we can see that with ERE, both SAC and SOP gain a significant performance improvement in all environments. Moreover, we also include results for IG+ERE and compare it with IG, SOP and SOP+ERE in Appendix G (figure 8). Some IG+ERE experiments are still running. We will update the figure once we have them.
> > >
> > > We have also fixed a list of typos and formatting issues.

---

> > > ### Author Response · Authors · 2019-11-15
> > > **Final revision finished**
> > >
> > > We have just finished the final revision. A summary of the changes has been posted separately for all reviewers. We would like to thank you again for your helpful feedback, they really helped us improve the quality of the paper. Thank you!

---

### Author Response · Authors · 2019-11-15
**Summary of changes**

We want to thank the reviewers again for their feedback, we have conducted a set of new experiments and now made a final revision. Here is a list of changes. We have also made some small fixes and reorganized some of the text.

Section 1 Introduction:
- Emphasize contributions of TD3: clipped double Q-learning, delayed policy update and target policy smoothing.
- Mention additional experiments with Inverting Gradients(IG).

Section 4 (Streamlined Off-Policy Algorithm)
- Remove the extra hyper-parameter beta.
- Include results comparing SAC, SOP, TD3+(TD3 plus normalization), and IG in figure 3.
- Add a description on the core idea of Inverting Gradients.

Section 5 (Non-uniform Sampling)
- Include results for SAC+ERE, comparing to SAC, SOP, and SOP+ERE in figure 4.

Related Work:
- Cite paper “Understanding the impact of entropy on policy optimization”.
- Cite several critique and reproducibility papers.

Appendix B:
- Add a concrete and thorough discussion of how we conducted hyper-parameter search.
- Provide hyper-parameters of additional experiments.

Appendix D:
- Add algorithmic and implementation details for Inverting Gradients.

Appendix E:
- Include algorithmic details and results for SOP+Exponential Sampling

Appendix G:
- Add discussion of implementation and computation complexity of new experiments

Appendix H:
- Include results for IG+ERE, and compare it with SOP+ERE, SOP and IG.
- Include results comparing TD3 with TD3+.

---

### Decision · Program_Chairs · 2019-12-19

**Decision:**

Reject

**Comment:**

The paper studies the role of entropy in maximum entropy RL, particularly in soft actor-critic, and proposes an action normalization scheme that leads to a new algorithm, called Streamlined Off-Policy (SOP), that does not maximize entropy, but retains or exceeds the performance of SAC. Independently from SOP, the paper also introduces Emphasizing Recent Experience (ERE) that samples minibatches from the replay buffer by prioritizing the most recent samples. After rounds of discussion and a revised version with added experiments, the reviewers viewed ERE as the main contribution, while had doubts regarding the claimed benefits of SOP. However, the paper is currently structured around SOP, and the effectiveness of ERE, which can be applied to any off-policy algorithm, is not properly studied. Therefore, I recommend rejection, but encourage the authors to revisit the work with an emphasis on ERE.